# Annexin A1 Is Associated with Adverse Clinical Outcomes in Patients with COVID-19

**DOI:** 10.3390/jcm11247486

**Published:** 2022-12-17

**Authors:** Matthias H. Busch, Sjoerd A. M. E. G. Timmermans, Joop P. Aendekerk, Renée Ysermans, Jean Amiral, Jan G. M. C. Damoiseaux, Chris P. Reutelingsperger, Pieter van Paassen

**Affiliations:** 1Department of Nephrology and Clinical Immunology, Maastricht University Medical Center, 6202AZ Maastricht, The Netherlands; 2Department of Biochemistry, Cardiovascular Research Institute Maastricht, Maastricht University, 6229HX Maastricht, The Netherlands; 3Scientific Hemostasis, 95130 Franconville, France; 4Central Diagnostic Laboratory, Maastricht University Medical Center, 6229HX Maastricht, The Netherlands

**Keywords:** COVID-19, Annexin A1, hyperinflammation, adverse outcomes

## Abstract

Severe coronavirus disease 2019 (COVID-19) is characterized by hyperinflammation, vascular damage, and hypercoagulability. Insufficient responses of Annexin A1 (AnxA1), a pro-resolving inhibitor of neutrophil infiltration and activation, might contribute to a severe course of the disease. We longitudinally evaluated AnxA1′s role in terms of inflammation, vascular damage, and clinical outcomes in a large prospective cohort of patients with COVID-19. AnxA1 was measured at presentation and during follow-up in the sera of 220 consecutive patients who presented at our hospital during the first wave. AnxA1 was significantly higher in the moderate and severe cases of COVID-19 compared to the healthy controls. Elevated AnxA1 was associated with markers of inflammation and endothelial damage. AnxA1 was significantly higher in patients with thrombotic events and ICU admission. Multivariable logistic regression indicated baseline AnxA1 (per ten units) as a predictor of thrombotic events. Linear mixed models predicted that AnxA1 tended to increase more steeply over time in patients without adverse events, with a statistically significant rise in patients without thrombotic events. These findings might reflect an insufficient increase in AnxA1 as a response to the excessive hyperinflammation in COVID-19. Future studies should evaluate whether hyperinflammation could be reduced through the administration of human recombinant AnxA1 or Ac2-26 peptide.

## 1. Introduction

Coronavirus disease 2019 (COVID-19) is caused by infection with severe acute respiratory syndrome coronavirus 2 (SARS-CoV-2), leading to a wide spectrum of clinical manifestations. Most patients remain asymptomatic, although life-threatening acute respiratory distress syndrome may occur against a background of hyperinflammation, vascular damage, and coagulopathy [1]. The innate immune system and, more specifically, neutrophils and neutrophil extracellular trap (NET) formation, are key factors in the development of severe COVID-19 [2].

Annexin A1 (AnxA1) is a member of the annexin superfamily and exerts its anti-inflammatory effects among a variety of cellular functions in humans through signaling via formyl peptide receptor 2 (FPR2) [3]. The active form of full-length AnxA1 (37-kDa) enhances neutrophil apoptosis and inhibits neutrophil infiltration and activation [4,5,6,7]. One may assume that deregulated homeostasis of AnxA1 can therefore play a role in the pathogenesis of severe COVID-19. Dexamethasone, an exogenous glucocorticoid upstream of AnxA1, [8] has, in fact, been shown to alter type I interferon active neutrophils in COVID-19 and improve outcomes for patients with severe COVID-19 [9,10]. Preliminary observations by Canacik et al. showed low levels of AnxA1 in patients with severe COVID-19 compared to those with moderate disease [11]. However, the sample size of the included patients was rather small. Moreover, longitudinal data on the changes in AnxA1 levels during the course of COVID-19 are lacking.

We studied the serum levels of AnxA1 at presentation and over time in a large and well-defined cohort of patients with COVID-19 to delineate AnxA1′s role in terms of inflammation, vascular damage, and clinical outcomes. 

## 2. Materials and Methods

### 2.1. Patient Population and Sampling

Consecutive patients with COVID-19 who presented during the first wave (21 March to 28 April 2020) at the emergency department of the Maastricht University Medical Center (MUMC), Maastricht, The Netherlands, were included, as previously reported [1]. Presenting patients without typical findings of COVID-19 according to computed tomography, such as diffuse ground glass opacities and/or bilateral consolidations, or according to reverse transcriptase-polymerase chain reaction of a nasopharyngeal swab and/or sputum (i.e., SARS-CoV-2 RNA with a cycle threshold value <40) were excluded. Patients were categorized as mild (not admitted to the hospital), moderate (admitted; requiring up to 5 L/min oxygen support), or severe (admitted; requiring more than 5 L/min oxygen support, or requiring invasive ventilation and/or COVID-19-related fatal courses). Blood samples were obtained at the time of presentation at the emergency department or intensive care unit (ICU) and whenever available during follow-up (i.e., every 5 (±2) days). After 30 min of clotting, serum tubes were centrifuged at 1885× *g* for 10 min at room temperature (RT). Citrated blood was immediately centrifuged at 2000× *g* for 10 min at RT. Subsequently, all samples were aliquoted and stored at −80 °C. 

### 2.2. Data Collection

Clinical characteristics and findings, as well as outcomes (i.e., disease severity, lengths of hospitalization and ventilation, thrombotic events, ICU admission, and 28-day in-hospital mortality), were retrieved from electronic patient files.

### 2.3. Annexin A1

AnxA1 was quantified in the serum via an enzyme-linked immunosorbent assay (ELISA; Scientific Hemostasis, Franconville, France). Briefly, rabbit anti-human recombinant AnxA1 pAbs (5 µg/mL diluted in a 0.05 M carbonate buffer at pH 9.6) were coated on microtiter plates overnight at RT and washed. Serum samples were diluted in a 1:5 ratio in 0.05 M PBS, 0.15 M NaCl, 1% BSA, and 0.1% Tween 20 (PBST); incubated for 1 h at RT; and washed. Next, peroxidase-labeled anti-AnxA1 pAbs (40 µg/mL diluted in PBST) were incubated for 1 h at RT, after which TMB/H2O2 substrate (Neogen, Lansin, MI) was added for 15 min at RT. The reaction was stopped with 0.5 M sulfuric acid, and AnxA1 was quantified in duplicate at 450 nm.

### 2.4. Soluble Complement 5a

Desarginated complement 5a (C5a) was quantified in plasma via ELISA (Quidel, San Diego, CA, USA), according to the manufacturer’s instructions.

### 2.5. Von Willebrand Factor Antigen

Von Willebrand factor antigen (vWF:Ag) was quantified in plasma as previously described [1].

### 2.6. Statistical Analysis

Depending on the normality and equal variances, continuous variables are indicated as median (interquartile range [IQR]) or mean (±standard deviation [SD]). Differences were assessed via Mann–Whitney U tests, unpaired sample *t* tests, Kruskal Wallis tests, or one-way ANOVA, as appropriate. Fisher’s exact test was used for categorical variables. Correlations between AnxA1 and inflammatory and endothelial markers were calculated using the Spearman’s rank correlation coefficient. To investigate associations between baseline AnxA1 (per ten units) and clinical outcomes (i.e., thrombotic events, ICU admission and 28-day in-hospital mortality), univariable and multivariable logistic regressions were performed. Depending on the clinical outcome, the multivariable analysis was adjusted for other factors influencing the clinical outcome. The results are presented as the odds ratio (OR) with the 95% confidence interval (CI) and area under the receiver operating characteristic curve (AUC). Linear mixed-effects models were run to investigate the effects of longitudinal AnxA1 on clinical outcomes. Statistics were performed using IBM SPSS Statistics (version 28) and GraphPad Prism (version 9). Statistical significance was assumed for *p*-values < 0.05.

## 3. Results

### 3.1. Patient Population

Overall, AnxA1 was assessed in 220 out of 228 (96%) patients with COVID-19; eight patients were excluded because of insufficient sampling. The baseline characteristics of the included patients stratified by disease severity are depicted in Table 1.

The median duration from the onset of symptoms to presentation was 7 (IQR, 5–12) days; 48 (22%), 68 (31%), and 104 (47%) patients presented with mild, moderate, and severe COVID-19, respectively. The comorbidities did not differ between the groups. In accordance with the national recommendations during the first wave, 135 (61%) were treated with hydroxychloroquine [12]. Patients were not routinely treated with immunosuppressive drugs, such as dexamethasone or interleukin 6 inhibition. Seven (3%) patients were treated with low-dose prednisolone, that is, <20 mg/d, because of comorbidities. 

### 3.2. Annexin A1

The AnxA1 levels are depicted in Figure 1. The mean and median AnxA1 levels were 16.8 (SD, ±8.5) ng/mL and 14.9 (IQR, 10.4–22.4) ng/mL, respectively, in 58 samples from healthy donors from the Central Diagnostic Laboratory, MUMC (donated before 2019). The AnxA1 reference value of ≤33.8 ng/mL was based on the mean (+2 × SD). Elevated levels of AnxA1 were found in 85 (39%) out of 220 patients at baseline; there were 11 (23%), 28 (41%), and 46 (44%) patients with mild, moderate, and severe disease, respectively. At presentation, AnxA1 was significantly higher in the moderate (30.1 (IQR, 16.0–42.0) ng/mL; *p* < 0.0001) and severe cases of COVID-19 (28.9 (IQR, 17.3–53.6) ng/mL; *p* < 0.0001) compared to the healthy controls (14.9 (IQR, 10.4–22.4) ng/mL). AnxA1 in the mild cases was also lower compared to the severe cases (20.4 (IQR, 11.8–32.2) ng/mL; *p* = 0.045). Notably, AnxA1 tended to increase over time in patients admitted to the hospital with COVID-19 (Figure 1B, Appendix A).

### 3.3. Annexin A1, Inflammation, and Endothelial Damage

The characteristics and markers of inflammation, as well as vWF:Ag, a marker of endothelial damage, in patients with normal and elevated AnxA1 at baseline are presented in Appendix A and Figure 2. Elevated levels of AnxA1 were associated with higher white blood cell counts (7.8 (IQR, 6.0–10.1) vs. 6.3 (IQR, 4.9–8.5) × 10^9^/L; *p* = 0.001), neutrophil counts (6.0 (IQR, 4.4–8.2) vs. 4.9 (IQR, 3.5–6.8) × 10^9^/L; *p* = 0.004); CRP (94 (IQR, 57–168) vs. 70 (IQR, 27–116) mg/L; *p* = 0.002); C5a (23.3 (IQR, 16.5–32.4) vs. 17.9 (IQR, 10.1–27.0) ng/mL; *p* = 0.007); and vWF:Ag (462 (IQR, 346–591) vs. 356 (IQR, 251–457)%; *p* < 0.0001) compared to the levels of AnxA1 within the normal range. Lymphocytes and the neutrophil-to-lymphocyte ratio (NLR) were not associated with AnxA1.

The Spearman’s rank correlation coefficients between AnxA1 and the markers of inflammation and endothelial damage at baseline were assessed (Appendix A). Only weak positive correlations between AnxA1 and leukocyte count (*r* = 0.309; *p* < 0.0001), neutrophil count (*r* = 0.297; *p* < 0.0001), CRP (*r* = 0.227; *p* = 0.001), and vWF:Ag (*r* = 0.315; *p* < 0.0001) were found. Lymphocytes (*r* = 0.160; *p* = 0.026), C5a (*r* = 0.190; *p* = 0.008), and NLR (*r* = 0.082; *p* = 0.457) also correlated poorly with AnxA1.

### 3.4. AnxA1 and Clinical Outcomes

The median number of days from the onset of symptoms to presentation was 9 (IQR, 7–14) days for patients with elevated AnxA1 at baseline and 7 (IQR, 5–10) days for patients with normal AnxA1 (*p* < 0.001). The AnxA1 levels appeared to be the highest in patients with thrombotic events and/or those admitted to the ICU (Figure 3).

The logistic regression analysis showed that increasing AnxA1 (per ten units) was associated with an increased risk of thrombotic events in patients with COVID-19 (OR 1.064 (95% CI 1.003–1.129); *p* = 0.040), as did the multivariable analysis (OR 1.067 (95% CI 1.002–1.135); *p* = 0.042) after adjustment for sex (Table 2, Appendix A). Increasing levels of AnxA1 (per ten units) did not show an increased risk of ICU admission (OR 1.052 (95% CI 0.997–1.111); *p* = 0.065) or 28-day in-hospital mortality (OR 1.050 (95% CI 0.982–1.122); *p* = 0.115).

Next, we ran linear mixed models to investigate differences in the dynamics of AnxA1 over time for several clinical outcomes (Figure 4 and Appendix A). The predicted AnxA1 at baseline tended to be higher in patients with thrombotic events and ICU admission, as well as in non-survivors; however, this was without statistical significance. Interestingly, AnxA1 tended to increase more steeply over time in patients without adverse events, with a statistically significant rise only in patients without thrombotic events (*p* = 0.048).

## 4. Discussion

In this study, we demonstrated that AnxA1 is significantly increased at presentation and during hospital admission in patients with moderate and severe COVID-19. This increase is associated with elevated markers of inflammation and endothelial damage. Clinically, elevated AnxA1 at presentation predicts an increased risk of thrombotic events in patients with COVID-19.

Elevated levels of AnxA1 may reflect the pro-inflammatory state of patients with moderate and severe COVID-19. AnxA1 is a pro-resolving mediator of inflammation and may counteract hyperinflammation. Indeed, AnxA1 was associated with markers of inflammation, including the potent anaphylatoxin C5a, and endothelial damage. AnxA1 exerts its anti-inflammatory properties by activating the FPR2 receptor on neutrophils, thereby increasing vascular neutrophil detachment, decreasing neutrophil transmigration to the target site, and improving apoptotic neutrophil phagocytosis [4,7].

AnxA1 did not differ between patients with moderate and severe COVID-19 at presentation and over time, despite the fact that several markers of inflammation, i.e., CRP and NLR, were higher in the severe cases. The capacity to secrete higher levels of AnxA1 to buffer the pro-inflammatory state was eventually exceeded in patients with severe COVID-19. Unfortunately, little is known about the feedback loops of extracellular AnxA1 secretion during excessive inflammation in humans. Based on our observations, potential therapeutic benefits for severe COVD-19 patients may arise by pharmacologically increasing FPR2 signaling through the administration of human recombinant AnxA1 or its FPR2-binding peptide (Ac2-26) [13,14].

Next, we found that higher levels of AnxA1 were associated with poor clinical outcomes. In particular, AnxA1 at presentation predicted the risk of thrombotic events after correction for sex. We also estimated longitudinal changes in AxnA1 in relation to clinical outcomes using linear mixed models. AnxA1 tended to increase more steeply over time in patients without adverse outcomes. The association between AnxA1 and adverse outcomes might reflect a reciprocal increase in AnxA1 in patients with excessive hyperinflammation. The notion that AnxA1 tends to increase less steeply over time in patients with adverse outcomes may reflect, once again, the saturated buffer capacity of AnxA1 in these patients. With respect to thrombotic events, it is unknown whether AnxA1 is directly involved in the regulation of the coagulation cascade and platelet activation or indirectly in thrombus formation through the reduction of neutrophil participation [15]. However, it is well-recognized that COVID-19-induced hypercoagulability is linked to excessive inflammation via neutrophil extracellular trap formation and complement activation, which triggers the intrinsic pathway of coagulation [1,16,17]. Increased neutrophil extracellular trap formation may, therefore, be accompanied by the release of intracellular AnxA1.

One study previously reported decreased AnxA1 in patients with severe COVID-19 compared to moderate cases and controls [11]. A cut-off value of 17.2 ng/mL for AnxA1 was suggested as a predictive biomarker for ICU admission, which was in contrast to our findings. There were differences between both cohorts. Moderate and severe cases were defined slightly differently in the studies. Moreover, the patients included in the cited study were predominantly female, particularly in the severe group (66%), which is uncommon for severe COVID-19 [18]. The actual impact of sex differences on AnxA1, however, is unknown. Our study included a significantly higher number of patients combined with longitudinal measurements, underpinning the robustness of our observations.

Our study has several limitations. First, the detection of AnxA1 in serum could lead to an overestimation of the actual AnxA1 levels, particularly in patients with higher neutrophil counts. AnxA1 is stored in granules of neutrophils [19] and, through centrifugation, intracellular AnxA1 could be released and artificially increase the serum fraction. However, we found only a weak correlation between AnxA1 and neutrophils in our cohort, making it unlikely that this effect would significantly affect our observations. Secondly, the sample size of the follow-up samples is limited.

In conclusion, we demonstrate that the pro-resolving mediator of inflammation, AnxA1, is increased in the sera of patients with moderate and severe COVID-19 and is associated with adverse clinical outcomes. We believe that these findings reflect an insufficient increase in AnxA1 as a response to the excessive hyperinflammation in these patients. Future studies should evaluate whether hyperinflammation in COVID-19 patients can be diminished by targeting FPR2 through the administration of human recombinant AnxA1 or Ac2-26 peptide.

## Figures and Tables

**Figure 1 jcm-11-07486-f001:**
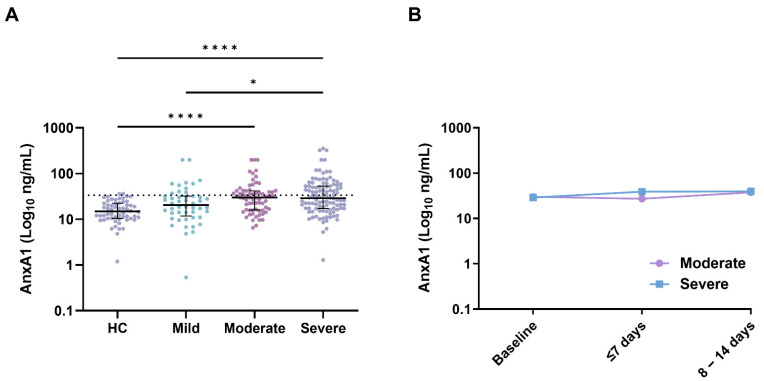
AnxA1 (Log10 ng/mL) at baseline in healthy controls and patients with mild, moderate, and severe COVID-19 (**A**) and during follow-up in patients with moderate and severe COVID-19 (**B**). Scatter plots with depicted medians and interquartile ranges for AnxA1. *p*-values were calculated using the Kruskal–Wallis test. * *p* < 0.05, **** *p* < 0.0001. Abbreviation: AnxA1—Annexin A1.

**Figure 2 jcm-11-07486-f002:**
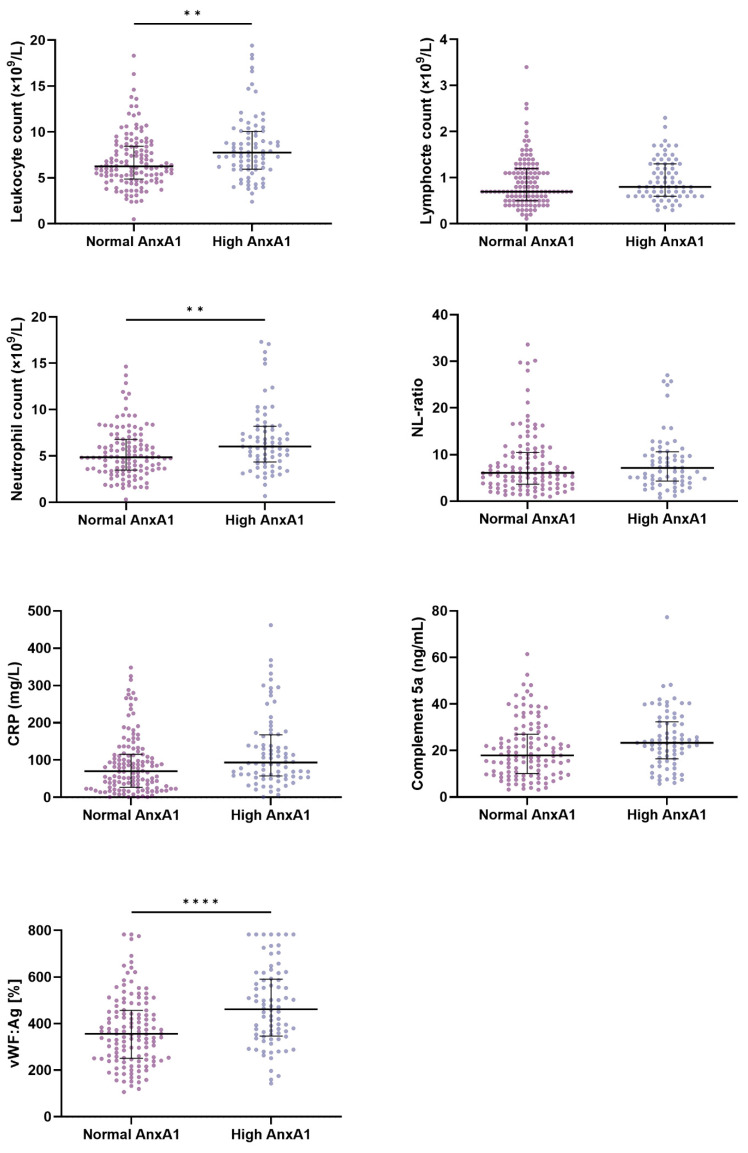
Markers of inflammation and endothelial activation in COVID-19 patients with normal and elevated AnxA1 levels (>33.8 ng/mL) at baseline. Scatter plots with depicted medians and interquartile ranges for normal and high AnxA1. *p*-values were calculated using the Mann–Whitney U test. ** *p* < 0.01, **** *p* < 0.0001. Abbreviations: AnxA1—Annexin A1; NL-ratio—neutrophil-to-lymphocyte ratio; CRP—C-reactive protein; vWF:Ag—von Willebrand factor antigen.

**Figure 3 jcm-11-07486-f003:**
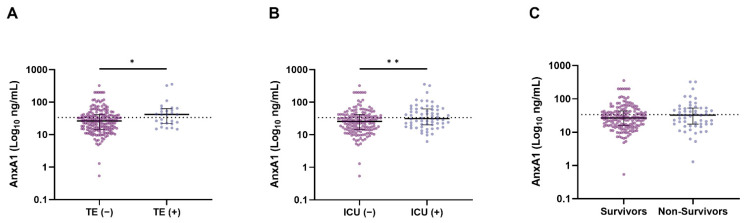
AnxA1 levels (Log10 ng/mL) in patients with COVID-19 with and without thrombotic events (TE) (**A**), ICU-admitted or not (**B**), and deceased or not during hospitalization (**C**). Scatter plots with depicted medians and interquartile ranges for AnxA1. *p*-values were calculated using the Mann–Whitney U test. * *p* < 0.05, ** *p* < 0.01. Abbreviations: AnxA1—Annexin A1; TE—thrombotic event; ICU—intensive care unit.

**Figure 4 jcm-11-07486-f004:**
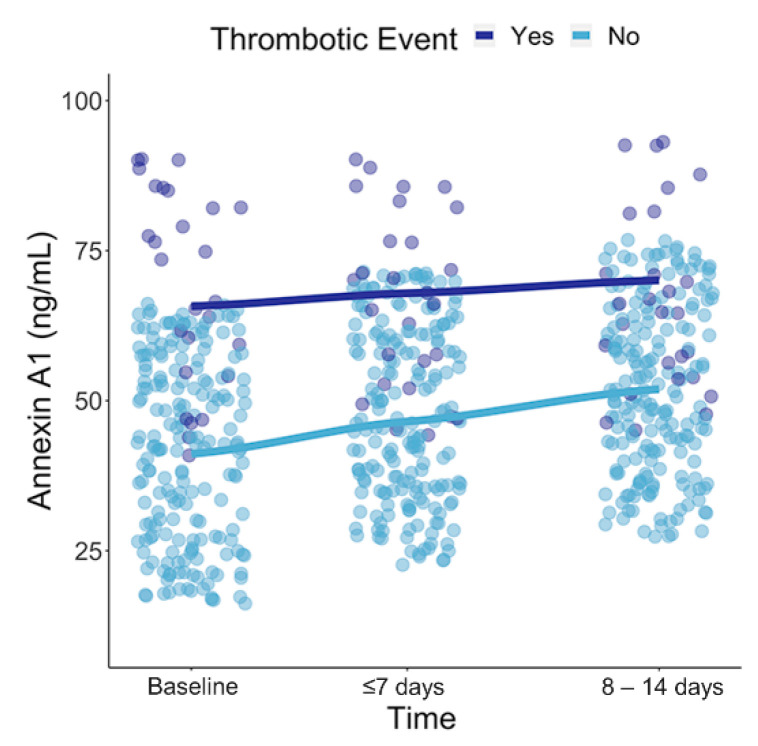
The predicted AnxA1 (ng/mL) at baseline and over time stratified by patients with and without thrombotic events using linear mixed models. Over time, AnxA1 levels increased significantly in patients without thrombotic events (*p* = 0.048).

**Table 1 jcm-11-07486-t001:** Baseline characteristics of 220 patients with COVID-19.

	Normal Range	Mild (*n* = 48)	Moderate (*n* = 68)	Severe (*n* = 104)	Overall *p*
M/F		25/23	43/25	76/28 ^†^	0.037
Age, yr.		62 (±16)	70 (±12) ^‡^	69 (±12) ^†^	0.002
Days from illness onset		7 (5−11)	7 (5−14)	7 (5−14)	0.963
SBP, mmHg		129 (±16)	137 (±22)	138 (±25)	0.068
DBP, mmHg		80 (±12)	79 (±12)	80 (±14)	0.885
Heart rate, bpm		90 (75−100)	90 (80−100)	95 (80−110) *^,†^	0.043
Body temperature, °C	≤37.9	37.7 (±0.9)	38.1 (±1.0) ^‡^	38.1 (±1.0) ^†^	0.021
Fever, *n* (%)		14 (30)	39 (58) ^‡^	53 (60) ^†^	0.002
Medical history					
Hypertension, *n* (%)		15 (31)	27 (40)	34 (33)	0.951
Diabetes, *n* (%)		11 (23)	11 (16)	25 (24)	0.673
CVA, *n* (%)		7 (15)	11 (16)	18 (17)	0.736
Cardiac disease, *n* (%)		13 (27)	24 (35)	31 (30)	0.899
COPD/asthma, *n* (%)		6 (13)	16 (24)	11 (11)	0.418
None, *n* (%)		12 (25)	15 (22)	27 (26)	0.804
Laboratory parameters					
Platelets, ×10^9^/L	130−350	195 (164−292)	202 (143−260)	211 (168−246)	0.795
Leukocytes, ×10^9^/L	3.5−11.0	6.0 (5.4−8.4)	6.3 (4.7−8.7)	7.4 (5.8−9.9) ^†^	0.009
Neutrophils, ×10^9^/L	1.4−7.7	4.6 (3.6−6.2)	5.0 (3.4−7.3)	5.9 (4.7−8.1) ^†^	<0.001
Lymphocytes, ×10^9^/L	1.1−4.0	1.1 (0.7−1.6)	0.8 (0.6−1.2) ^‡^	0.7 (0.5−1.1) ^†^	0.002
NL-ratio		4.6 (2.8−6.6)	6.0 (3.9–9.0) ^‡^	8.7 (5.3–12.2) *^,†^	<0.001
AST, U/L	<35	38 (27−56)	49 (37−64) ^‡^	55 (40−80) ^†^	<0.001
LDH, U/L	<250	256 (205−339)	328 (266−451) ^‡^	451 (358−595) *^,†^	<0.001
Serum creatinine, µmol/L	60−115	83 (62−106)	88 (75−119)	91 (73−120)	0.254
Albumin, g/L	32.0−47.0	34 (31−38)	33 (30−36)	29 (26−32) *^,†^	<0.001
CRP, mg/L	<10	56 (16−95)	66 (39−123)	98 (54−174) *^,†^	<0.001
C5a, ng/mL	≤21.1	15.3 (9.0−25.4)	21.8 (16.2−30.7) ^‡^	21.8 (10.8−30.7) ^†^	0.025
High C5a, n/N		25/41	50/56 ^‡^	70/95 *	0.005
AnxA1, ng/mL	≤33.8	20.4 (11.8−32.2)	30.1 (16.0−42.0)	28.9 (17.3−53.6) ^†^	0.025
High AnxA1, *n* (%)		11 (23)	28 (41) ^‡^	46 (44) ^†^	0.023

C5a was not measured in all patients because of insufficient sampling. Continuous variables are presented as mean (±standard deviation) or median (interquartile range) as appropriate. Differences between groups were analyzed via unpaired sample *t* test, Mann–Whitney U test, or Kruskal Wallis test. Differences in categorical variables were analyzed via chi square test or Fisher’s exact test when appropriate; significant differences between patient groups: severe versus * moderate or ^†^ mild disease; moderate versus ^‡^ mild disease. Abbreviations: SBP—systolic blood pressure; DBP—diastolic blood pressure; CVA—cerebrovascular accident; NL-ratio—neutrophil-to-lymphocyte ratio; COPD—chronic obstructive pulmonary disease; AST—aspartate transaminase; LDH—lactate dehydrogenase; CRP—C-reactive protein; C5a—complement 5a; AnxA1—Annexin A1.

**Table 2 jcm-11-07486-t002:** Binomial logistic regression was used to evaluate the effects of increasing AnxA1 (per ten units) alone (univariable) and together with other predictors (multivariable) on the likelihood of thrombotic events, ICU admission, and 28-day mortality, respectively.

Univariable	OR (95% CI)	*p*-Value	AUC (95% CI)
Thrombotic events	1.064 (1.003–1.129)	0.040	0.638 (0.535–0.741)
ICU admission	1.052 (0.997–1.111)	0.065	0.611 (0.531–0.691)
28-day mortality	1.050 (0.982–1.122)	0.115	0.627 (0.514–0.740)
**Multivariable ***	**OR (95% CI)**	***p*-value**	**AUC (95% CI)**
Thrombotic events *	1.067 (1.002–1.135)	0.042	0.729 (0.629–0.829)
ICU admission ^#^	1.043 (0.967–1.125)	0.280	0.759 (0.690–0.828)
28-day mortality ^±^	0.993 (0.923–1.068)	0.851	0.765 (0.690–0.839)

* Adjusted for sex. ^#^ Adjusted for CRP (mg/L), sex, hypertension, and cardiac disease as comorbidities. *^±^* Adjusted for CRP (mg/L), age in years, and diabetes as comorbidities. Abbreviation: ICU—intensive care unit.

## Data Availability

The original data from this study can be obtained upon reasonable request from the corresponding author.

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
