# Peer review of "Annexin A1 Is Associated with Adverse Clinical Outcomes in Patients with COVID-19"

_jcm, 2022, doi:10.3390/jcm11247486_

Round 1
Reviewer 1 Report
Authors have evaluated AnxA1’s role in terms of inflammation, vascular damage, and clinical outcomes in a cohort of patients with COVID-19.
Introduction
1) Page 1, lines 43-45. Please provide the reference "Deregulated homeostasis of AnxA1 may therefore play a role in the pathogenesis of severe COVID-19"
2) lines 52-54, It would be helpful if authors can explain more what dynamics of AnxA1 exactly means here.
Material and methods
1) There is no information on inclusion and exclusion criteria of patients
2) Patients recruitment has been done only for a period of nearly one month in early 2020 which is indeed a long time ago, and its a big limitation
3) It would be helpful to provide more information on blood sampling
4) line 71, The "28 days in hospital" is not clear , authors need to clarify
Result
1) lines 120-122, Please provide the reference "According to national recommendations"
2) lines 122, patients refer to which group of patients (mild, moderate, sever)
3) line 165, It would be helpful if authors clarify with more information on onset of symptoms to presentation
Discussion
1) Authors need to recheck and provide the reference for the missed ones.
2) Authors have mentioned that Elevated AnxA1 was associated with markers of inflammation and endothelial damage but in lines 242-247 it has been said that they have observed a week correlation between higher AnxA1 and neutrophil count, It would be helpful if you can clarify more this correlation and if you have found a weak correlation does that mean your AnxA1 is not overestimated?
Author Response
Reviewer 1
Comments and Suggestions for Authors
Authors have evaluated AnxA1’s role in terms of inflammation, vascular damage, and clinical outcomes in a cohort of patients with COVID-19.
Introduction
1) Page 1, lines 43-45. Please provide the reference "Deregulated homeostasis of AnxA1 may therefore play a role in the pathogenesis of severe COVID-19"
We thank the reviewer for the valuable comments on our manuscript. The line is an assumption or hypothesis and is not supported by scientific data, we used “may” to indicate this. For a better clarification we rephrased the line into “One may assume that a deregulated homeostasis of AnxA1 can therefore play a role in the pathogenesis of severe COVID-19.”
2) Lines 52-54, It would be helpful if authors can explain more what dynamics of AnxA1 exactly means here.
We changed the line into “We studied serum levels of AnxA1 at presentation and over time (…)” to improve the understanding what “dynamics” means in our manuscript.
Material and methods
1) There is no information on inclusion and exclusion criteria of patients.
We updated the patient population and sampling section accordingly.
2) Patients recruitment has been done only for a period of nearly one month in early 2020 which is indeed a long time ago, and its a big limitation.
We agree that patient recruitment took place a long time ago when the COVID-19 pandemic begun. This may be a limitation because treatment strategies and SARS-CoV-2 variants have changed. In turn, immunosuppressants like glucocorticosteroids were not prescribed in our cohort on a regular basis back then. This allows to study the actual effects of SARS-CoV-2 on Annexin A1 biology in patients with COVID-19 because glucocorticosteroids are known to interfere with Annexin A1 expression and increase cellular surface localization.1
3) It would be helpful to provide more information on blood sampling.
We updated the section accordingly.
4) line 71, The "28 days in hospital" is not clear , authors need to clarify.
We changed “28 days in-hospital” into “28 days in-hospital mortality”.
Result
1) lines 120-122, Please provide the reference "According to national recommendations".
According to the journals manual, we added an archived website linked to the Dutch guidelines for the treatment of patients with COVID-19 at that time (archived on 22 April 2020). Unfortunately, there is only a Dutch translation available.
2) lines 122, patients refer to which group of patients (mild, moderate, severe).
Line 122 summarizes the total number of included patients (n=220). The subgroups (i.e., mild, moderate, and severe) are further described on page 4 and in Table 1. We have changed the line into: “Overall, AnxA1 was assessed in 220 out of 228 (96%) patients with COVID-19; 8 patients were excluded because of insufficient sampling. The baseline characteristics of the included patients stratified by disease severity are depicted in Table 1.”
3) line 165, It would be helpful if authors clarify with more information on onset of symptoms to presentation.
Patients with elevated AnxA1 had ~ 2 days longer symptoms of COVID-19 (i.e., fever, cough, dyspnea, malaise, gastro-intestinal symptoms, […]) at home before presenting at the emergency department compared to patients with normal AnxA1. We included in the supplementary an additional table with baseline characteristics of patients with normal and elevated AnxA1 at presentation (Supplementary S2).
Discussion
1) Authors need to recheck and provide the reference for the missed ones.
We have rechecked the discussion part and updated missing references whenever applicable.
2) Authors have mentioned that Elevated AnxA1 was associated with markers of inflammation and endothelial damage but in lines 242-247 it has been said that they have observed a week correlation between higher AnxA1 and neutrophil count, It would be helpful if you can clarify more this correlation and if you have found a weak correlation does that mean your AnxA1 is not overestimated?
Indeed, neutrophils contain high levels of AnxA1 intracellularly. The release of AnxA1 from neutrophils during the processing of the samples could thereby lead to an overestimation of the actual AnxA1 levels measured in the serum of the patients. Since we did not find a strong correlation between AnxA1 levels and neutrophils, this potential confounder is not likely to be present in our study. We pointed this out in the limitations section of the discussion.
REFERENCES
1 Perretti, M. & Dalli, J. Exploiting the Annexin A1 pathway for the development of novel anti-inflammatory therapeutics. Br J Pharmacol 158, 936-946 (2009).
Reviewer 2 Report
Comments:
1) Must improve the language of the manuscript
2) Add more experimental data
3) Include recent references to the Introduction part and improve this section
4) Discussion should be improved with appropriate discussion of the relevant studies.
Author Response
Comments and Suggestions for Authors
1) Must improve the language of the manuscript.
We thank the reviewer for his comments. We used the English editing service offered form the journal to improve the language of the manuscript.
2) Add more experimental data
Unfortunately, we cannot add more experimental data based on this rather unspecific comment. Though, we have included extra supplemental data and figures based on the comments from reviewer 1 and 3.
3) Include recent references to the Introduction part and improve this section and 4) Discussion should be improved with appropriate discussion of the relevant studies.
Since the reviewer is not specific about what exactly to improve in the introduction and discussion, we have improved the manuscript based on the reviewers opinion 1 and 3. We have also done a new search on published original articles addressing AnxA1 blood levels during active SARS-CoV-2 infection. We found one additional original article not cited in our manuscript.1 Importantly, the article is published in Turkish only; we were not able to review the study and therefore did not include it in our manuscript.
REFERENCES
1 Ural, O. et al. [Evaluation of Annexin-1 (ANXA-1), Annexin-2 (ANXA-2) and Bone Morphogenetic Protein-7 (BMP-7) Serum Levels in Patients Followed Up With A Diagnosis of COVID-19]. Mikrobiyol Bul 56, 25-35 (2022).
Reviewer 3 Report
The list of comments and suggestions:
- The origin of AnxA1 reference value of ≤33.8 ng/mL is unclear. According to Canacik et al. 2021 the vales of AnxA1 in control subjects were 25 (18-39) ng/ml (median (interquartile ranges)). The values of AnxA1 for healthy donors used in the study should be provided as median and interquartile ranges, for easier comparison with COVID-19 subjects. Also more information on the control group should be provided. Is normal range of AnxA1 is known?
- Regarding statistical analysis – what statistical test was used to test the significance of AnxA1 differences over time?
- Table 1 – Body temperature normal range is given as >37.9, probably a mistake.
- The results presented in Table 3 are unclear. The rise of AnxA1 in patients with and without TE should be shown on a graph.
- Significant correlations between AnxA1 and markers of inflammation and endothelial damage should be presented on graphs, if not in the paper then in supplementary materials.
Author Response
The origin of AnxA1 reference value of ≤33.8 ng/mL is unclear. According to Canacik et al. 2021 the vales of AnxA1 in control subjects were 25 (18-39) ng/ml (median (interquartile ranges)). The values of AnxA1 for healthy donors used in the study should be provided as median and interquartile ranges, for easier comparison with COVID-19 subjects. Also more information on the control group should be provided. Is normal range of AnxA1 is known?
We thank the reviewer for his comments and feedback.
We calculated the reference value of ≤33.8 ng/mL based the mean + 2 x standard deviation; that is 16.8 + (2 x 8.5) ng/mL. The reference value in terms of median + interquartile range would be 14.9 (IQR, 10.4-22.4) ng/mL. This information has been updated in the manuscript. We decided to use the mean + 2 x standard deviation because a) this is the generally more accepted method for calculating a reference value in a cohort of healthy donors and b) the AnxA1 serum levels in the donor cohort passed normality (D’Agostino & Pearson omnibus normality test; P = 0.131).
AnxA1 levels were measured in serum samples (n=58) obtained before 2019 from the healthy donor biobank of the Central Diagnostic Lab of the Maastricht University Medical Center.
There are no normal ranges available for AnxA1. AnxA1 levels vary depending on the Kit and whether serum or plasma is used. For example, Liu et al. reported a mean of 1.03 (SD, 0.60) ng/mL in peripheral blood of 20 healthy controls (Bio-Rad 550, US).1 Lee et al. measured AnxA1 in plasma of 25 healthy controls and found AnxA1 levels of 0.312 ± 0.078 ng/mL (R&D System, Minneapolis, MN, USA).2 Finally, Purvis et al. found in another study AnxA1 levels between 20-50 ng/mL in 20 healthy donors (home-made ELISA).3
Regarding statistical analysis – what statistical test was used to test the significance of AnxA1 differences over time?
We used the Mann Whitney U test for testing the significance between moderate and severe patients for each time-point (i.e., baseline, ≤ 7 days, and 8 -14 days). We added this information in the Supplementary S1.
To test the significance within the different groups over time, we used the Friedman test. Retrospectively, however, the Friedman test should not have been used in this case because there are missing data on time-point ≤ 7 days and 8 -14 days. An alternative would be the Skillings-Mack test, but there are most likely too many missing data to reliably test the significance with data imputation. Of note, missing data over time occurred most frequently because patients were discharged, were transferred to another hospital or died. We removed the second table from the Supplementary S1.
Table 1 – Body temperature normal range is given as >37.9, probably a mistake.
We corrected this mistake into ≤37.9.
The results presented in Table 3 are unclear. The rise of AnxA1 in patients with and without TE should be shown on a graph.
We updated the manuscript by adding Figure 4 showing the TE data and moving Table 3 to Supplementary S5.
Significant correlations between AnxA1 and markers of inflammation and endothelial damage should be presented on graphs, if not in the paper then in supplementary materials.
We added a correlation matrix in Supplementary S3.
REFERENCES
1 Liu, Q., Ding, M., Sun, J. & Zhu, Z. Identification of ANXA1 as a diagnostic and lymphatic metastasis factor in colorectal cancer. J Tre Bio Res 1, 1-4 (2018).
2 Lee, S. H. et al. Annexin A1 in plasma from patients with bronchial asthma: its association with lung function. BMC Pulm Med 18, 1 (2018).
3 Purvis, G. S. D. et al. Annexin A1 attenuates microvascular complications through restoration of Akt signalling in a murine model of type 1 diabetes. Diabetologia 61, 482-495 (2018).
Round 2
Reviewer 1 Report
I would like to thank authors for replying my comments.
Reviewer 2 Report
I have gone through the revised manuscript and recommend to Accept this manuscript.